# Photo-Methionine, Azidohomoalanine and Homopropargylglycine Are Incorporated into Newly Synthesized Proteins at Different Rates and Differentially Affect the Growth and Protein Expression Levels of Auxotrophic and Prototrophic *E. coli* in Minimal Medium

**DOI:** 10.3390/ijms241411779

**Published:** 2023-07-22

**Authors:** Tomas Jecmen, Roman Tuzhilkin, Miroslav Sulc

**Affiliations:** Department of Biochemistry, Faculty of Science, Charles University, Albertov 2030, 128 43 Prague, Czech Republic

**Keywords:** non-canonical amino-acid-containing proteins, bio-orthogonal amino acid global substitution, azidohomoalanine, homopropargylglycine, photo-methionine, *E. coli*

## Abstract

Residue-specific incorporation of non-canonical amino acids (ncAAs) introduces bio-orthogonal functionalities into proteins. As such, this technique is applied in protein characterization and quantification. Here, we studied protein expression with three methionine analogs, namely photo-methionine (pMet), azidohomoalanine (Aha) and homopropargylglycine (Hpg), in prototrophic *E. coli* BL-21 and auxotrophic *E. coli* B834 to maximize ncAA content, thereby assessing the effect of ncAAs on bacterial growth and the expression of cytochrome b_5_ (b_5_M46), green fluorescence protein (MBP-GFP) and phage shock protein A. In auxotrophic *E. coli*, ncAA incorporation ranged from 50 to 70% for pMet and reached approximately 50% for Aha, after 26 h expression, with medium and low expression levels of MBP-GFP and b_5_M46, respectively. In the prototrophic strain, by contrast, the protein expression levels were higher, albeit with a sharp decrease in the ncAA content after the first hours of expression. Similar expression levels and 70–80% incorporation rates were achieved in both bacterial strains with Hpg. Our findings provide guidance for expressing proteins with a high content of ncAAs, highlight pitfalls in determining the levels of methionine replacement by ncAAs by MALDI-TOF mass spectrometry and indicate a possible systematic bias in metabolic labeling techniques using Aha or Hpg.

## 1. Introduction

Bioorthogonal amino acids are non-canonical amino acids (ncAAs) with functional groups that differ from those naturally found in canonical amino acids (cAAs). ncAAs are biologically inert and do not interfere with normal cellular processes when incorporated into proteins. Among ncAAs, three methionine analogs have garnered significant attention. A diazirine functional group containing photo-methionine (pMet) was used as a UV-light-activated zero-length crosslinker in structural proteomics [1,2], whereas azidohomoalanine (Aha) and homopropargylglycine (Hpg), which contain azide and alkyne moiety, respectively, were used for selective conjugation with various other molecules, such as affinity labels and fluorescent dyes [3,4]. 

Incorporating these ncAAs into proteins has become a powerful tool with numerous applications in basic and applied research. Such applications include the visualization and quantitation of nascent proteins in live cells, organisms and ecosystems [5,6,7]; the development of new engineered proteins serving as biocatalysts, therapeutics or protein materials with unique properties [8,9] and the study of protein structure, function and interactions [10,11]. Methionine residues also stand out as candidates for substitution because they are common ligands of metal atoms in the active sites of electron transfer proteins, affecting the redox properties and reactivity of these proteins, and can be found along the pathways whereby electrons flow between the active site of a protein and an electron donor or acceptor. Modifying protein scaffolds of these pathways by introducing methionine analogs with delocalized electrons distributed on their side chains may increase the efficiency and rate of electron transfer. As a case in point, the rates of a reaction mediated by cytochrome P450 2B4 significantly increase in the presence of cytochrome b_5_ mutants with pMet introduced to two distinct positions within the putative cytochrome P450 binding site, as shown in our previous study [12].

The high fidelity of the genetic code is under stringent control of amino acid-tRNA synthetases (aaRSs). These enzymes charge individual cAAs onto cognate tRNAs and can also activate synthetic ncAAs structurally similar to a cAA at varying rates [13]. In particular, methionyl-tRNA synthetase can be used to introduce ncAAs into proteins, given its substrate promiscuity. An ncAA competes with a canonical counterpart for attachment to a specific tRNA by the corresponding aaRS. The ratio between the resulting ncAA-tRNA and cAA-tRNA is dictated by the difference in the catalytic efficiencies (k_cat_/K_M_) of the aaRS for the respective amino acids and by the concentration of the amino acids. In the next step, amino acyl-tRNAs bind to elongation factor EF-Tu, forming a ternary complex that specifically interacts with ribosomes based on the tRNA anticodon sequence. Although both ncAA and cAA are incorporated into growing polypeptide chains in this manner, the efficiency of this process may be low for non-native substrates due to tighter or weaker interactions with EF-Tu [14].

The minimum substitution rate required for different applications varies with the ncAA. Even replacing as little as 6% methionine residues by Aha suffices for experiments derived from bioorthogonal noncanonical amino acid tagging (BONCAT). However, uniform methionine substitution at different protein positions and minimal impact on cell metabolism by free forms of ncAAs is more important than a high level of cAA substitution for the technique [15]. We have previously photo-crosslinked two purified interaction partners—cytochromes b_5_ and P450—and characterized the resulting covalent protein complex with only 35% methionine residues in cytochrome b_5_ replaced with pMet [16]. Partial substitution of methionine by its analogs has been achieved in several prokaryotic and eukaryotic cells under different experimental conditions [5,6,7,16]. Nevertheless, for experiments requiring large amounts of protein with substitution levels well above 50%, the protocol must be fine-tuned to the specifics of the organism expressing the protein and of the incorporated ncAA. Such optimization is required to determine or alter electron transport pathways through model proteins, including cytochrome b_5_ and azurin, using ncAAs introduced at key locations within the protein [17].

ncAA-based approaches primarily aim at achieving an ncAA incorporation efficiency high enough for the intended application while maintaining a satisfactory level of protein expression and minimizing the perturbation of cell metabolism by free forms of the ncAA. The last factor is especially relevant for determining the overall proteosynthetic activity in ecosystems and for identifying and quantifying newly synthesized proteins in cells or organisms. Yet, despite considerable advances in the last decade, ncAA incorporation into proteins still poses challenges due to several factors, not all of which can be easily adjusted during the experiments. Among them, neither the cellular uptake mechanism and rate of different ncAAs nor the cellular concentration of ncAA can be fully controlled because they may vary in time and space after administration to experimental organisms. Additionally, methionine can be at most partly depleted in cells (especially in prototrophic organisms that synthesize methionine); its residual amounts compete for methionyl-tRNA synthetase (MetRS) and are subsequently incorporated into proteins. The rate of ncAA incorporation also depends on the efficiency of translation and is determined by the properties of the nascent polypeptide chain. In general, experimental expression protocols should be optimized for each protein to achieve a high substitution rate of cAA for ncAA. 

In this study, we compared the expression of three different proteins, namely recombinant cytochrome b_5_ mutant (b_5_M46), recombinant maltose-binding protein fused with a green fluorescent protein (MBP-GFP) and phage shock protein A (pspA), containing three distinct bio-orthogonal methionine analogs in prototrophic and auxotrophic *E. coli*. Cytochrome b_5_ is a small (15 kDa), membrane-bound electron carrier with a regulatory role in a mixed-function oxygenase system, which has long been studied in our group [18]. Here, we expressed its selected mutants with a high content of ncAAs. b_5_M46 contains a single methionine site on a readily ionizable tryptic peptide (at position 46), which enables us to determine the relative abundance of different amino acids at the methionine site with high accuracy by MS. In turn, MBP-GFP is a larger (74.5 kDa) protein with 12 methionine sites. This protein can be easily quantified due to its fluorescence. Lastly, pspA is a 25.5-kDa protein with seven methionine sites that plays a key role in bacterial survival under nutritional stress. In this study, pspA caught our attention because its expression was significantly elevated under specific experimental conditions. Our findings may provide an experimental paradigm for producing other ncAA-containing proteins and optimizing experimental conditions for monitoring the protein turnover or overall metabolic activity of microorganisms.

## 2. Results

In this study, we compared the growth of *E. coli* BL-21 Gold (DE3) prototroph and *E. coli* B834 (DE3) methionine auxotroph in methionine-deficient mineral medium supplemented with pMet, Aha, and Hpg analogs, the rate of protein expression under these conditions, and the efficiency of methionine replacement by these ncAAs in newly synthesized proteins. 

### 2.1. Prototrophic E. coli Grow in Mineral Medium Supplemented with pMet and Aha but Not Hpg

Cultures of both *E. coli* strains grown overnight in LB medium were used as inocula for batches of minimal media supplemented with either methionine or one of its non-canonical analogs, and the subsequent increase in biomass was monitored by measuring the optical density. OD_600_ values were fitted to growth curves (Figure 1), and selected growth characteristics were calculated (Table 1).

Prototrophic *E. coli* BL-21 (Figure 1A) showed similar growth curves in the presence of pMet and Aha, as confirmed by their growth rate and maximum population size. Both OD_600_ at 5% of the maximum population size (a point followed by a steep increase in optical density in all cultures whose OD_600_ increased above the noise level during the experiment) and OD_600_ at the stationary growth phase (a point where the growth rate declined to 1% of the maximum value) were reached after longer periods in media supplemented with pMet or Aha than in medium supplemented with methionine. But while the maximum growth rate was ~40% lower in the presence of pMet and Aha than in medium with methionine, the maximum population size was, rather surprisingly, ~10% higher. The excess of supplemented methionine might have inhibited growth by entering other metabolic pathways and disrupting their metabolite balance. This effect was not observed in experiments with ncAAs, given their limited ability to enter physiological processes other than proteosynthesis. 

In the presence of Hpg, *E. coli* BL-21 culture did not exceed OD_600_ ~0.1 throughout the experiment, thus indicating its strong effect on cell metabolism. Accordingly, either the effect of Hpg cannot be offset by the increase in methionine concentration by de novo synthesis, unlike the effect of the other two ncAAs, or methionine concentration remains low because biosynthesis is blocked. This effect can be deduced from the differences in the growth of prototrophic and auxotrophic *E. coli* between media supplemented with pMet (Figure 1D) and Aha (Figure 1E). By contrast, in media supplemented with Hpg, both strains showed similar levels of growth inhibition (Figure 1F). 

Auxotrophic *E. coli* B834 growth (Figure 1B) was substantially inhibited by all three ncAAs, but after approximately 16 h of cultivation, OD_600_ started to increase gradually in minimal medium with Aha. Nevertheless, all cultures remained viable throughout the experiment, as confirmed by their inoculation into fresh LB medium, which restored their growth. Both bacterial strains grew to the same maximum OD_600_ in minimal medium with methionine but slightly differed in the other metrics, with the auxotrophic strain reaching 5% of the maximum OD_600_ earlier and entering the stationary phase later than the prototrophic strain. In line with these differences, the maximum growth rate of the auxotrophic strain was also ~25% lower (Figure 1C), as observed in the LB medium (Appendix A). 

Based on these results, pMet and Aha have similar but not identical effects on bacterial growth when methionine is unavailable because a small amount of this amino acid is critical for cell growth and division. Prototrophic *E. coli* can compensate for this deficiency by methionine biosynthesis, but cells must adapt and build up the necessary enzymatic apparatus for this purpose, as shown by the delayed growth. In sharp contrast, supplementing the medium with Hpg had a strong effect on prototrophic *E. coli* cells, indicating that either methionine biosynthesis was inhibited or the cellular metabolism was severely affected in some other way because their growth did not differ significantly from that of the auxotrophic strain in media containing ncAAs.

### 2.2. Prototrophic E. coli Undergo a Second Growth Phase, Whereas Auxotrophic E. coli Enter a Late Stationary Phase When Depleting Methionine Accumulated in Cells by Transfer from Nutritionally Rich to Mineral Medium with ncAA

Cultures of both *E. coli* strains transformed with a plasmid encoding either b_5_M46 or MBP-GFP were grown in LB medium until reaching a late exponential phase (OD_600_ ~0.6) when they were washed and transferred into the minimal medium supplemented with one of the ncAAs before inducing the expression of the respective protein. Subsequently, the OD_600_ of the cultures (Figure 2), the amount of synthesized b_5_M46 (Figure 3) or MBP-GFP (Figure 4), and the degree of methionine substitution by the ncAAs (Figure 5 and Figure 6) were monitored as well.

Both auxotrophic and prototrophic *E. coli* strains grew immediately after being transferred to minimal media supplemented with one of the ncAAs. Subsequently, the growth rate gradually decreased over several hours under all tested conditions (Figure 2). However, in the presence of pMet and Aha, the prototrophic strain then entered a second phase of diauxic growth, whereas the auxotrophic strain showed a slow population decline typical of the death phase of bacterial growth. A similar pattern, albeit less pronounced, was observed in bacteria expressing MBP-GFP in an Hpg-containing medium but not in those expressing b_5_M46, which showed no diauxic growth throughout the experiment. 

The auxotrophic and prototrophic *E. coli* strains displayed similar growth patterns and higher population sizes in medium supplemented with pMet or Aha than in medium supplemented with Hpg (Figure 2A–F), in line with the previous experiment (Figure 1D–F). However, the initial growth phase observed under all conditions in the second experiment (Figure 2A–F) was detected only when supplementing the minimal medium with methionine (Figure 1C), not with pMet, Aha or Hpg (Figure 1D–F). Nevertheless, in the second experiment, the starting amount of bacteria in the minimal medium, which is a source of methionine contained in the biomass, was 10 times higher than in the first experiment. Therefore, the amount of methionine available to the bacteria in the initial growth phase seemed to be sufficient to meet their demands, and accordingly, the prototrophic strain did not synthesize any extra methionine; otherwise, the growth of the prototrophic strain would have surpassed that of the auxotrophic strain. After residual methionine depletion, all cultures entered the stationary phase, during which the cellular machinery for methionine biosynthesis was activated in prototrophic bacteria grown in a minimal medium supplemented with pMet or Aha, thereby restoring their growth.

### 2.3. Protein Expression Rates Are Higher in the Presence of pMet Than in the Presence of Aha and in Prototrophic Than in Auxotrophic E. coli

To compare the number of proteins expressed by the auxotrophic and prototrophic *E. coli* strains, we sampled the bacterial suspensions during expression, applied them to SDS-PAGE, and estimated the amount of the proteins under investigation at each time point based on the intensities of corresponding bands. Although the prototrophic culture grew at similar rates in the presence of pMet and Aha (Figure 2A,B), the rate of b_5_M46 expression was significantly higher in the former than in the latter (Figure 3A,B). The amount of protein synthesized in the first 2 h of expression in the presence of pMet matched the amount obtained in 4–6 h in the presence of Aha. This trend was also observed at the later time points. Protein expression in the presence of Hpg (Figure 3C) was generally lower than in the presence of the other two ncAAs due to the lower number of bacteria synthesizing the protein throughout the experiment.

The same total amount of protein obtained after 21 h of expression in the presence of Hpg was produced after only 2–6 h when supplementing the medium with one of the other two ncAAs. MBP-GFP expression showed similar trends in SDS-PAGE (Appendix A), in line with fluorescence measurements (Figure 4D), which provided a more accurate quantification of this protein. 

The auxotrophic strain showed a lower overall growth with pMet and Aha than the prototrophic strain and, consequently, a lower protein expression. The same amount of MBP-GFP expressed after 21 h in the auxotrophic strain was reached after only ~12 h in the prototrophic strain. The auxotrophic strain also required a period more than 10 times longer to synthesize the same amount of b_5_M46 as that synthesized by the prototrophic strain within 2 h. Furthermore, the total amount of b_5_M46 expressed by the auxotrophic strain was lower in the presence of pMet than in the presence of Aha (Figure 3D,E), partly due to the lower OD_600_ of the culture grown in a medium supplemented with pMet (Figure 2A,B). Given the same number of bacteria, the total amount of protein synthesized might have been slightly higher in the presence of pMet than in the presence of Aha, as observed in the MBP-GFP expression, but SDS-PAGE is not sufficiently accurate to support such a conclusion.

In contrast to the other two ncAAs, the similar growth of the two *E. coli* strains in a medium supplemented with Hpg (Figure 2C,F) was also associated with almost identical expression rates of the same protein (Figure 3C,F and Figure 4C). Both b_5_M46 and MBP-GFP were expressed by the prototrophic strain at a lower rate with Hpg than with the other two ncAAs. In the auxotrophic strain, MBP-GFP expression was also ~1.6 and ~4.3 times slower in the presence of Hpg than in the presence of Aha and pMet, respectively (Figure 4E). Conversely, the b_5_M46 expression rate of the auxotrophic strain with Hpg was the highest of the three ncAAs (Figure 3D–F).

One bacterial protein, identified by mass fingerprinting as pspA (Appendix A), also showed elevated levels in the presence of Hpg in both *E. coli* strains (Figure 3C,F) and in the presence of Aha in *E. coli* B834 (Figure 3E). In all cases, its expression followed the trend of b_5_M46 synthesis, albeit at lower total levels when expressed with Hpg (similarly in both *E. coli* strains) and higher levels when expressed with Aha. Increased pspA expression indicates that the bacteria were under nutritional or another type of stress [19], which resulted in the activation of the metabolic pathway associated with survival involving this protein. Although we only found increased expression of this protein, the levels of other proteins that respond to adverse conditions most likely also changed concurrently but were not detected by SDS-PAGE due to the low sensitivity of this method. Nevertheless, the pspA levels did not change during MBP-GFP expression under any experimental conditions, nor during b_5_M46 expression with pMet in both *E. coli* strains or with Aha in the *E. coli* prototrophic strain. This perturbation of the bacterial proteome might have derived from the cumulative effect of supplementation with a specific ncAA and an additional stimulus, such as overexpression of a specific protein or even from a subtle variation in a step of the protocol that is difficult to reproduce accurately (e.g., washing and transferring the culture to minimal medium, during which bacteria can be disturbed to varying degrees). Such a perturbation of the bacterial metabolome has been previously shown in prototrophic *E. coli* K12. In these cells, treatment with Aha and Hpg changed metabolite levels only minimally, but the changes became more pronounced when these treatments were combined with heat stress [20].

In summary, protein expression rates in the presence of pMet exceeded those in the presence of Aha and were generally higher in prototrophic than in auxotrophic *E. coli*, regardless of the protein. By contrast, the expression rates were similar in the two *E. coli* strains grown in a medium supplemented with Hpg and higher than in a medium supplemented with pMet or Aha for poorly expressed proteins but lower for readily expressed proteins. Under some conditions, shock response protein A was overexpressed in the cells.

### 2.4. Incorporation of Non-Canonical Amino Acids

The levels of methionine substitution, analyzed by MS, were mostly similar between proteins, with a higher incorporation of pMet and Aha in the auxotrophic *E. coli*. In turn, the prototrophic strain appeared to be more suitable for protein expression in a medium supplemented with Hpg. However, when determining the ncAA incorporation rate based on MALDI-TOF data, we identified a systematic bias in our pMet and Aha results.

For each time point, ncAA and bacterial strain, the proteins of interest were proteolytically digested, and the peptide mixtures were analyzed by mass spectrometry (MS). Mass spectra were searched for b_5_M46, MBP-GFP and pspA tryptic peptides containing one or two methionine sites, showing intense signals (Table 2). Variants of these peptides with individual methionine sites occupied by amino acids listed in Table 3 were found in the spectra, and the intensities of their signals were used to determine the aggregate incorporation rate of an ncAA and its derivatives. Depending on the ncAA, its derivatives were products of bacterial metabolism (either before or after incorporation into proteins), arose during sample processing or were formed during the ionization. All samples were characterized using a rapid and straightforward MALDI-TOF MS approach, and selected samples were additionally characterized by LC-ESI-qTOF MS, which is better suited for analyzing peptides with higher molecular weights and more accurately quantifying low-abundance species.

In a previous study [1], we have shown that pMet incorporation by prototrophic *E. coli* peaks immediately after inducing cytochrome b_5_ expression in a minimal medium and drops sharply within the first 4 h without significantly changing subsequently. Here, we were able to determine whether a significant increase or decrease in ncAA incorporation occurred in this initial phase under specific conditions, albeit without reliably determining the pattern of the change because the temporal resolution is lower in this initial phase and the determination of ncAA incorporation is more biased in this time window due to the low amount of proteins available for analysis. After the fourth hour, the trends in ncAA incorporation rates remained almost linear under all conditions (Figure 5 and Figure 6).

Prototrophic bacterial cultures showing similar growth but different recombinant expression dynamics in the presence of pMet and Aha also incorporated both ncAAs in a different manner. The overall trend was similar; the levels of ncAA incorporation did not change further after an initial decline. However, in the presence of pMet, the levels of incorporation in both b_5_M46 and MBP-GFP plateaued at 20–30% (Figure 5A). Conversely, in the presence of Aha, the level of incorporation in MBP-GFP, which was synthesized at a lower rate, ranged from 30 to 40% at later time points, whereas the level of incorporation in b_5_M46 sharply decreased from an initial level of approximately 30% to less than 10% after 2 h (Figure 5B). Both proteins were synthesized at lower amounts in the presence of Hpg than in the presence of the other two ncAAs, but at a much higher rate of ncAA incorporation, producing these proteins with a substitution level of ~80% (Figure 5C). The incorporation of Aha and Hpg calculated independently for the two MBP-GFP peptides differed by up to 15–20% at the initial time points and gradually converged to less than 10% once more protein was synthesized and subsequently available for MS analysis.

The proportion of ncAA and methionine in both proteins recombinantly expressed by the auxotrophic *E. coli* strain was determined by MS and LC-MS for all three ncAAs, except for Hpg incorporation into b_5_M46, whose levels were determined using MS only. Regardless of the approach, methionine analog incorporation into both proteins increased during the first four hours of expression for all ncAAs. This trend contrasted with the decrease in pMet and Aha incorporation observed in the prototrophic strain but matched the trend for Hpg. After this time point, the levels of ncAA incorporation did not change significantly (Figure 6A–D,F) except for Hpg incorporation into MBP-GFP, which continued to increase slightly, as during the expression in the prototrophic strain (Figure 6G).

We compared methionine substitution by the respective ncAAs at the methionine sites of two MBP-GFP peptides–SAMPEGYIQER (P1) and MEYNYNAHNVYIMTDK (P2)–in the time range of 4–26 h (Table 4). Regardless of the analytical approach, the peptides did not significantly differ in their levels of methionine substitution as the absolute value of the difference in the mean values for P1 and P2 (Δ Ave _(P1-P2)_) was close to the standard deviation from the mean value for both peptides separately (S.D. _(P1/2)_) or combined (S.D. _(total)_). For example, Δ Ave _(P1-P2)_ value 2.7 was lower than the respective S.D. _(P1/2)_ values of 5.4 and 3.6, and the S.D. _(total)_ value of 4.8 in the MS analysis of pMet incorporation into MBP-GFP.

The results of ncAA substitution determined for the same sample by MS and LC-MS analysis were inconsistent, with the values differing by up to ~20 percentage points (Δ Ave _(MS-LCMS)_). Nevertheless, the trends were consistent, except for the standard deviations (S.D. _(P1/2)_, S.D. _(total)_), which were lower in LC-MS. The difference in substitution rates between MALDI-TOF MS and LC-MS most likely derived from a systematic bias because the values of ncAA incorporation into b_5_M46 and MBP-GFP peptides were systematically lower for pMet and higher for Aha in the former than those determined using the more accurate LC-MS approach. 

The peptides with pMet D1 derivative were detected after MALDI ionization because the diazirine functional group of pMet is photolytically cleaved during its irradiation with a UV laser. We hypothesized that a fraction of the laser energy consumed in the photolysis lowered the ionization yield of the peptide with this ncAA, whereas the ionization yield of the methionine-containing peptide did not change. pMet photolysis is avoided in LC-MS, and therefore, the aforementioned bias was minimized. For Aha, some amount was converted into one of the derivatives (Aha D1–3) by bacteria or during the sample preparation procedure (see Table 3). Consequently, a mixture of the peptide forms with different abundances was detected using this method. Underrepresented forms of peptides may render the calculation of the Aha substitution rate inaccurate and, accordingly, higher if calculated based on MALDI-TOF data. 

An alternative explanation for both pMet and Aha may also be differences in the ionizability of their individual forms assessed by MALDI and ESI. The systematic differences in substitution determined by MS and LC-MS (Δ Ave _(MS-LCMS)_) fell within the standard deviation of the measurements (S.D. _(total)_) for Hpg and, as such, were deemed nonsignificant. An example of the MS and LC-MS data used to determine the rate of ncAA incorporation into b_5_M46 expressed for 26 h in *E. coli* B834 is shown in Appendix A. The *m*/*z* signal intensities of MALDI-TOF spectra and AUCs of LC-MS analyses that were used to calculate the levels of methionine substitution by ncAA at different times are summarized in Appendix A.

After b_5_M46 expression induction with Aha and Hpg in *E. coli* B834, the bacterial protein pspA, which is involved in adaptation to different types of stress, was overexpressed, mirroring b_5_M46 (Figure 3E,F). Hpg substitution determined by MALDI-TOF was ~70% in both b_5_M46 and pspA (Figure 6F,H). In contrast to b_5_M46 (Figure 6C), the levels of Aha substitution in pspA slightly increased over time (Figure 6E). This protein is constitutively expressed and, as such, was already present in bacterial cells before their transfer to the minimal medium in which the pspA expression rate increased considerably. Accordingly, a gradual increase in the levels of Aha substitution may be explained by a constant rate of ncAA incorporation but with a decreasing proportion of the previously synthesized protein (exclusively with methionine) in the total amount of pspA. Furthermore, the actual substitution levels should be 10–20 percentage points lower than those determined by MS (Figure 6F), reflecting the difference in substitution rates determined by MS and LC-MS for b_5_M46 and MBP-GFP (Figure 6C,D). Considering this difference, we assume that both Aha and Hpg incorporation levels into simultaneously synthesized proteins remain similar on a long timescale.

## 3. Discussion

In a previous study, we prepared cytochrome b_5_ with pMet content high enough for its use in light-induced protein cross-linking [16]. We noted that some mutants (e.g., b_5_M46) containing this ncAA increased the rate of the reaction catalyzed by cytochrome P450 2B4, a redox partner to which cytochrome b_5_ donates an electron needed for the reaction. We attributed this observation to a faster electron transfer between the proteins involving the diazirine group of pMet. For a more in-depth analysis, we required cytochrome b_5_ mutants with a methionine site substituted with pMet at a higher frequency and with two other ncAAs, namely Aha and Hpg, which, similarly to pMet, can be introduced into the protein sequence during recombinant expression in the bacterial host [13]. To our knowledge, these analogs have not yet been used to study electron transfer, but they are widely used in a number of other applications involving click chemistry [21].

In particular, the extent to which Aha and Hpg affect the metabolism of the cell under study is crucial for techniques examining cellular protein turnover. In this regard, the two ncAAs have been compared in a number of studies to determine which is better suited for a given experimental setup and the concentration at which the respective ncAA does not adversely affect cells. In two pioneering studies [3,22], cell viability was the main indicator of cellular impairment, but a more in-depth analysis is currently performed to characterize individual changes induced by ncAAs both qualitatively and quantitatively to avoid introducing selective bias into the results.

In one such approach, Aha and Hpg minimally affected the bacterial metabolome, but changes in the levels of various cellular metabolites resulting from a reaction to a separate stress factor (heat shock) were amplified when the bacteria were also simultaneously exposed to the ncAAs [20]. These findings are in line with our observations as the expression of b_5_M46, but not MBP-GFP also increased the expression of pspA protein in auxotrophic *E. coli* in the presence of Aha, and in both *E. coli* strains, in the presence of Hpg. Either proteosynthesis of a protein or a slight deviation from the experimental procedure might have generated a stimulus similar to that mentioned above, whose combined effect with the presence of the ncAAs has led to the change observed in the bacteria at the proteome level.

The impact of different Aha and Hpg concentrations on a prototrophic *E. coli* (strain DSM 103246) has also been reported in a recent study monitoring bacterial growth in a microwell plate format in conjunction with BONCAT staining and subsequent flow cytometry [22]. Under experimental conditions similar to those applied in our study, the results of the inhibition caused by Aha and Hpg and their effect on protein expression were also in line with our findings. This microwell plate approach makes it possible to concomitantly screen a wide range of experimental conditions but is unsuitable for continuous sampling, which is required to determine the ncAA incorporation rate in a time-dependent manner. Moreover, the conditions used to assess results on a micro-scale often differ from those that provide the same results on a large scale, required to produce large amounts of protein, due to the inherent variability associated with various physical and biological factors. 

Tivendale [7] showed that the concentrations of methionine (and other intermediates of its biosynthesis pathway) increased in *Arabidopsis thaliana* prototrophic cells grown for 10 h in a minimal medium in the presence of Aha but not Hpg. When growing prototrophic *E. coli* expressing proteins under similar experimental conditions, we also observed the onset of a second phase of diauxic growth after 10–15 h of culture in the presence of Aha (and pMet), but not Hpg. We attribute this decrease in the growth rate to a gradual depletion of residual methionine available to the bacteria. The subsequent resumption of growth after the lag phase, when methionine becomes available due to the activation of methionine biosynthetic pathways under some conditions, is supported by these findings. Additionally, the similarity between these results implies that some of our general conclusions may be applied to protein expression in eukaryotic cells. Nevertheless, considerable caution is required as plant cell growth is inhibited to a higher degree by Aha than by Hpg [7], in contrast to the results in prokaryotic cells observed in our experiments and previously published in the literature [23].

As discussed above, our findings are in line with the recent literature and complement the results of a different approach. Furthermore, they also bring a unique perspective in several aspects. Most studies merely compare growth inhibition or other parameters due to Aha and Hpg and relate them to a reference organism growing in the presence of methionine. Such an approach fails to distinguish the effects of the presence of an ncAA from those of the absence of methionine. pMet incorporates into proteins at a rate similar to that of Aha and Hpg, with minimal effects on cell metabolism. Therefore, pMet is a more appropriate reference than methionine. 

Based on the characteristics of diauxic growth under specific experimental conditions, the time required to deplete residual methionine after transferring a bacterial culture to a minimal medium may also be estimated. Supplementing the medium with ncAA and inducing expression after this delay should increase the rate of ncAA incorporation into the target protein. Our results further indicate that the rate of ncAA substitution changes noticeably shortly after transferring cells into a minimal medium supplemented with ncAA before stabilizing after a few hours. Thus, for prototrophic *E. coli* in the presence of Aha, the amount of a protein whose synthesis starts earlier in the initial phase may be overvalued relative to the amount of protein whose expression starts later as a consequence of the sharp decrease in the rate of Aha incorporation in this initial phase. 

Conversely, in the presence of Hpg and in auxotrophic *E. coli* grown in the presence of Aha, the amount of protein synthesized earlier may be undervalued for analogous reasons. This correlation is particularly important for the interpretation of BONCAT experiments, where only a short initial period (half an hour to a few hours) of proteosynthesis is tracked. Lastly, our results show that the ncAAs were similarly incorporated into different positions within a single protein and between different proteins simultaneously expressed in the bacterial cultures.

## 4. Materials and Methods

### 4.1. Materials and Reagents

Amino acids (except methionine analogs), Ampicillin, Coomassie Brilliant Blue R250, Formic Acid, Iodoacetamide, Tris(2-carboxyethyl)phosphine (TCEP), Thiamine and Trifluoroacetic Acid were purchased from Sigma-Aldrich (St. Louis, MO, USA); 1,4-Dithiothreitol (DTT) and Isopropyl β-D-1-thiogalactopyranoside (IPTG) were purchased from Carl Roth (Karlsruhe, Germany); LC-MS grade Acetonitrile and Water were purchased from Merck (Darmstadt, Germany); Trypsin was purchased from Promega (Madison, WI, USA); 2-Cyano-4-hydroxycinnamic Acid was purchased from Bruker (Bremen, Germany); L-Photo-Methionine (pMet) was purchased from Thermo Scientific (Rockford, IL, USA); L-Azidohomoalanine (Aha) and L-Homopropargylglycine (Hpg) were purchased from Jena Bioscience (Jena, Germany); 5-Aminolevulinic Acid was purchased from Chemos (Mělník, Czech Republic); the other chemicals were purchased from Lachema (Brno, Czech Republic). All chemicals were of analytical grade or better.

Bacterial strains *E. coli* BL-21(DE3) Gold and *E. coli* B834(DE3) were purchased from Merck (Darmstadt, Germany). The pET plasmid encoding mutant cytochrome b_5_ (lac promoter, ampicillin resistance, T7 driven) was cloned as described previously [24], and the plasmid encoding the fusion MBP-GFP protein (tac promoter, ampicillin resistance, T7 driven) was derived from the pMAL-c5X-MAP vector and kindly provided by Ondřej Vaněk (Faculty of Science, Charles University, Prague, Czech Republic).

### 4.2. Bacterial Growth

*E. coli* strains were transformed with one of the plasmids. Bacterial cultures grown overnight in Lysogeny broth (LB) medium supplemented with ampicillin (100 mg/L) were used to inoculate 25 mL of M9 minimal medium (the composition specified in Appendix A) supplemented with methionine or one of its analogs (1 mM) in 50-mL tubes to OD_600_ ~0.04. Two cultures were grown in parallel for each medium at 37 °C on a shaker (180 rpm), and their OD_600_ was measured in a cuvette with a 1-cm optical path on a DS-11 FX+ spectrophotometer (DeNovix, Wilmington, DE, USA) repeatedly. Growth curves were fitted to the measured OD_600_, and growth metrics were determined using Growthcurver [25]. The parameters t_5%K_ and t_1%r_ were derived from the fitted growth curves.

### 4.3. Protein Expression

Overnight *E. coli* cultures transformed with one of the plasmids were used to inoculate 150 mL of fresh LB medium in a 500-mL Erlenmeyer flask, and the cultures were grown at 37 °C on a shaker (180 rpm) to OD_600_ ~0.8. Each bacterial suspension was aliquoted into three 50 mL tubes, pelleted by centrifugation for 5 min at 5000× *g*, washed thoroughly with 15 mL of sterile phosphate-buffered saline (PBS) per tube and pelleted again under the same conditions. The pellet in each tube was resuspended in M9 medium supplemented with one of methionine analogs, and the resulting suspension was then transferred into two 50 mL tubes and appropriately diluted to OD_600_ 0.6 in 25 mL of bacterial suspension containing a methionine analog (1 mM). Additionally, M9 medium for cytochrome b_5_ expression was supplemented with 5-aminolevulinic acid (0.5 mM). The cultures were incubated for 30 min at 30 °C on a shaker (160 rpm) before inducing protein expression by adding 0.6 mM IPTG. The OD_600_ of the bacterial suspensions was measured, and a 100 µL aliquot was collected for electrophoresis at selected time points. Fluorescence (excitation: 442–497 nm, emission: 514–567 nm) of 10-fold diluted cultures were also measured during the expression of MBP-GFP on a DS-11+ with FX Fluorometer Module (DeNovix, Wilmington, DE, USA).

### 4.4. Electrophoresis and Protein Digestion

Bacterial suspension aliquots for electrophoresis were pelleted by centrifugation for 2 min at 9000× *g*, and the pellets were resuspended in 100 µL of sterile water and sonicated on a UP100H ultrasonic homogenizer (Hielscher, Teltow, Germany) for 10 s with amplitude set to 40%. The samples were mixed with 5× concentrated sample buffer containing 500 mM dithiothreitol as a reducing agent, incubated 5 min at 95 °C and loaded onto a self-cast discontinuous sodium dodecyl sulfate–polyacrylamide gel (10 × 6 × 0.1 cm, 10 wells). Separation was performed in Tris-glycine running buffer at a maximum voltage of 200 V and a maximum current of 25 mA. The gels were stained with 3 mM Coomassie Brilliant Blue R250 in a methanol–water–acetic acid mixture (45:45:10) and destained in an ethanol-water-acetic acid mixture (55:35:10).

The samples were processed for MS as described previously [26]. Briefly, the protein zones of interest were excised and destained in 50 mM 4-ethylmorpholine buffer (pH 8.1) in 50% (*v*/*v*) acetonitrile, all cysteine residues in the proteins were reduced in 30 mM TCEP at 60 °C for 20 min and alkylated in 30 mM iodoacetamide at room temperature in the dark for 1 h, and the proteins were in-gel digested by trypsin (4 ng/µL) in 100 mM ethylmorpholine buffer (pH 8.5) in 10% (*v*/*v*) acetonitrile at 37 °C for 12 h. A small portion of the resulting peptide mixture (~1.5%) was analyzed by MALDI-TOF MS, whereas the remaining supernatant was separated, and the residual peptides were extracted from the gel into 25 µL of 0.5% formic acid in 80% acetonitrile. Both supernatants were mixed, lyophilized, resuspended in 200 µL of 0.5% formic acid and desalted on a Peptide MicroTrap™ column (Michrom Bioresources, Auburn, CA, USA) according to manufacturer’s instructions. The samples were lyophilized and resuspended in 10 µL of 0.1% formic acid prior to LC-MS analysis.

### 4.5. MALDI-TOF Analysis

Each peptide mixture (0.5 µL) was spotted onto a MALDI target and overlaid with 0.5 µL of matrix (5 mg/mL 2-cyano-4-hydroxycinnamic acid dissolved in 50% (*v*/*v*) acetonitrile containing 0.1% (*v*/*v*) trifluoroacetic acid) using the dry droplet technique. Positive ion mass spectra were acquired over a 600–3000 *m*/*z* range on an Autoflex mass spectrometer (Bruker, Bremen, Germany) and manually evaluated using flexAnalysis 3.3 software (Bruker Daltonics, Bremen, Germany).

For known proteins, theoretical masses of the peptides containing one or more methionine residues or methionine analogs were generated using GPMAW software 7.1 [27]. Unknown proteins were identified by peptide mass fingerprinting [28]. Monoisotopic peptide masses were matched against the SWISSPROT non-redundant database; mass tolerance was set to 20 ppm, proteins were restricted to *E. coli*, allowing one missed cleavage and cysteine alkylation and methionine oxidation were set as fixed and variable modifications, respectively. The identity of unknown proteins was confirmed by tandem mass spectrometry with CID fragmentation of selected peptide precursors.

### 4.6. LC-MS/MS Analysis

Chromatographic separation was performed on an Agilent 1290 Infinity II LC System with a binary pump (Agilent Technologies, Waldbronn, Germany) interfaced with maXis Q-TOF mass spectrometer equipped with an ESI source (Bruker Daltonics, Bremen, Germany). A portion of the peptide sample (8 µL) was loaded onto a reversed-phase column (Zorbax RRHD Eclipse Plus C18, 2.1 × 100 mm, 1.8 µm; Agilent Technologies, Santa Clara, CA, USA) heated to 40 °C and separated using the following mobile phase gradient: solvent A, 0.1% (*v*/*v*) formic acid; solvent B, 0.1% (*v*/*v*) formic acid in acetonitrile; gradient (in % of buffer B), 3–6% (*v*/*v*) over 1 min, 6–30% (*v*/*v*) over 20 min, 30–80% (*v*/*v*) over 5 min; flow rate, 250 µL/min. MS1 spectra were acquired in positive mode over a 150–2200 *m*/*z* range at a 1 Hz spectral rate, and precursors with a minimal intensity of 5 × 10^4^ and preferred charge states 2–5 were selected for CID fragmentation. Other parameters of the MS method are provided in Appendix A. Data processing was performed using DataAnalysis 4.4 software (Bruker Daltonics, Bremen, Germany). Monoisotopic *m*/*z* values at charge states 1–5 for individual peptides with methionine, oxidized methionine and different forms of ncAA were calculated, generating extracted-ion chromatograms (EICs) for these masses. The identity of the peptides was confirmed by characteristic fragments in tandem mass spectra.

### 4.7. Estimation of the Frequency of Methionine Analog Incorporation

The percentage of methionine substituted by ncAA in the proteins was calculated for individual peptides. For MALDI-TOF data, the baseline was subtracted from the spectra using a built-in function of flexAnalysis software, and the relative intensities of the most abundant isotope of a peptide containing methionine or ncAA were used in the calculations; for LC-MS data, the chromatogram baseline was subtracted from individual EIC traces, setting the flatness parameter to 0.8 using a built-in function of DataAnalysis software, and the relative areas under curve were determined for peptides with methionine or ncAA.

The percentage of methionine sites substituted by an ncAA or any of its derivatives was determined for peptides with a single methionine site as
(1)$$Substitution\left(Pep\right)=\frac{\sum Pep\left(ncAA_{1}\right)}{\sum Pep\left(total\right)}\times 100\;\left[\%\right]$$
and for peptides with two methionine sites as
(2)$$Substitution\left(Pep\right)=\frac{\sum Pep\left(ncAA_{2}\right)+\frac{1}{2}\sum Pep\left(ncAA_{1}\right)}{\sum Pep\left(total\right)}\times 100\;\left[\%\right]$$
where ΣPep(ncAA_1_), ΣPep(ncAA_2_) and ΣPep(total) represent the sum of intensities (MALDI-TOF) or the sum of AUCs (RP-ESI-qTOF) assigned to a peptide containing one of the ncAA variants at exactly one position, two positions and any number of positions, respectively.

## 5. Conclusions

Only prototrophic *E. coli* BL-21 are suitable for protein expression in minimal medium supplemented with pMet and Aha. However, under these conditions, these bacteria express proteins with only a low ncAA content. A sufficient protein amount can be obtained when the bacterial host (both prototrophic and auxotrophic for methionine) is precultured in a rich medium and then transferred to a minimal medium with ncAA before inducing expression. Accordingly, the expression should be induced after the residual methionine available to the bacteria after their transfer from the rich medium is depleted; otherwise, this residual methionine will interfere with ncAA incorporation. This delay can reach the time required for prototrophic *E. coli* to enter the plateau of diauxic growth, which, under these conditions, is approximately 10 h in medium supplemented with pMet and Aha and 15 h in medium supplemented with Hpg. 

For both pMet and Aha, auxotrophic *E. coli* provides higher incorporation rates and lower protein yields than prototrophic *E. coli*. *E. coli* B834 shows the highest yield of readily expressed MBP-GFP in the presence of pMet, with the incorporation rate exceeding 70%, whereas b_5_M46 is only expressed at low levels, with the incorporation rate slightly above 50% under the same conditions. The incorporation of Aha into both proteins reaches ~50% in auxotrophic *E. coli*, with only a slightly lower level of incorporation into MBP-GFP in the prototrophic strain after two hours of expression. Thus, short-time expression in a prototrophic strain should be prioritized to reach a much higher protein yield with a slightly lower Aha content. Alternatively, the levels of Aha substitution may be maximized at the expense of protein yield in auxotrophic *E. coli*.

## Figures and Tables

**Figure 1 ijms-24-11779-f001:**
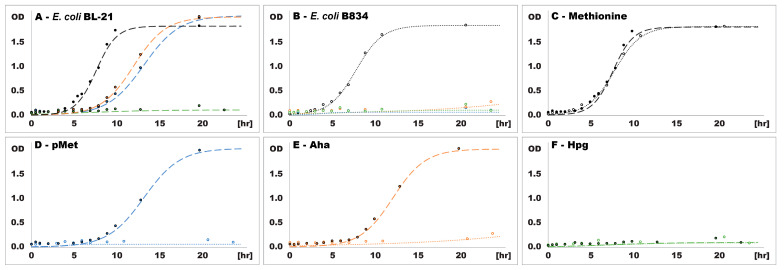
Bacterial growth in the presence of methionine and ncAAs. Growth curves of *E. coli* BL-21 (**A**) and *E. coli* B834 (**B**) in the presence of individual amino acids and of both *E. coli* strains in the presence of methionine (**C**), pMet (**D**), Aha (**E**) and Hpg (**F**). OD—optical density, *E. coli* BL-21 (dashed line), *E. coli* B834 (dotted line), methionine (black), pMet (blue), Aha (orange), Hpg (green).

**Figure 2 ijms-24-11779-f002:**
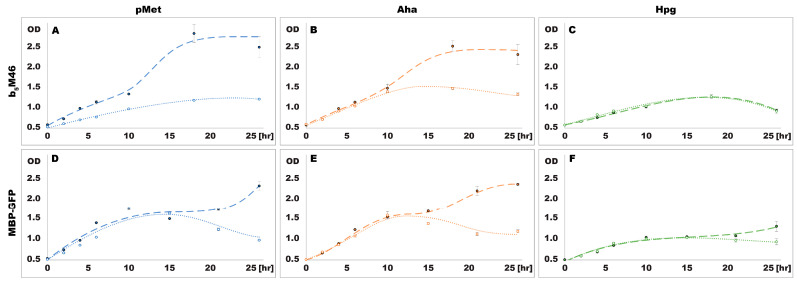
Bacterial growth during protein expression in the presence of pMet (**A**,**D**), Aha (**B**,**E**) and Hpg (**C**,**F**). OD—optical density, *E. coli* BL-21 (dashed line), *E. coli* B834 (dotted line).

**Figure 3 ijms-24-11779-f003:**
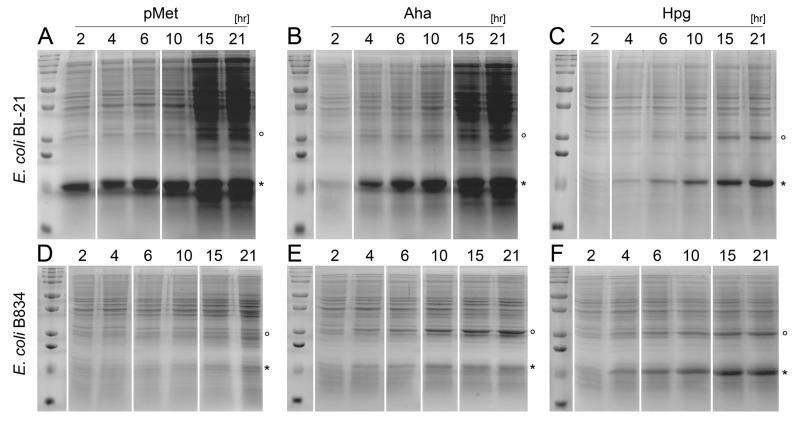
b_5_M46 expression levels in the presence of ncAAs. The same volumes of bacterial suspension collected during cultivation at indicated time points were separated on 15% SDS-PAGE stained by Coomassie Brilliant Blue R250 stain. Some samples were duplicated on the gels and have been removed from the figure for clarity at the locations indicated by white lines. Expression in *E. coli* BL-21 (**A**–**C**) and B834 (**D**–**F**) in the presence of pMet (**A**,**D**), Aha (**B**,**E**) and Hpg (**C**,**F**); marked proteins: * b_5_M46, ° pspA.

**Figure 4 ijms-24-11779-f004:**
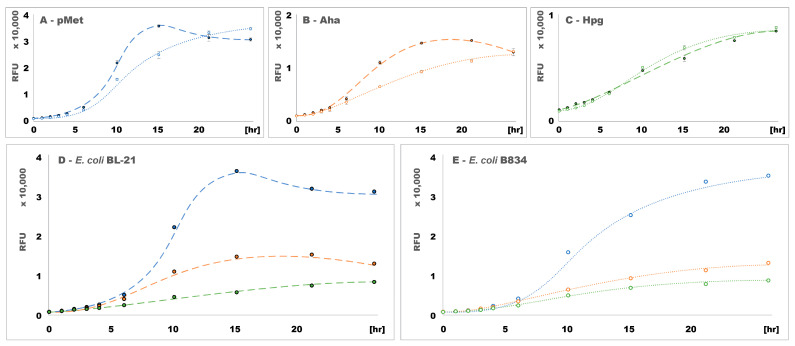
Fluorescence of bacterial cultures expressing MBP-GFP in the presence of the ncAAs. Excitation: 442–497 nm, emission: 514–567 nm. RFU—relative fluorescence unit; differential effect of pMet (**A**, blue line), Aha (**B**, orange line) and Hpg (**C**, green line) on the expression in both *E. coli* strains; differential effects of the ncAAs on *E. coli* BL-21 (**D**, dashed line) and B834 (**E**, dotted line) expression.

**Figure 5 ijms-24-11779-f005:**
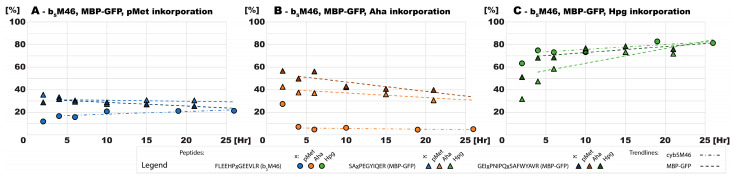
Aggregated incorporation of the ncAAs and their derivatives into the proteins in *E. coli* BL-21. Peptides originating from different proteins are distinguished by their specific mark, and different ncAAs are color-coded, as indicated in the figure captions. The trend lines were obtained by linearly interleaving the values at time points 4–26 h.

**Figure 6 ijms-24-11779-f006:**
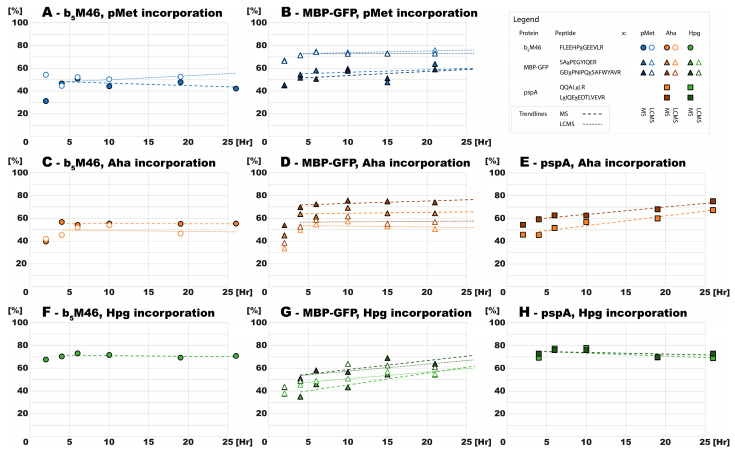
Aggregated incorporation of the ncAAs and their derivatives into the proteins in *E. coli* B834. Peptides originating from the different proteins are distinguished by their specific mark, and different ncAAs are color-coded, as indicated in the figure captions. The fill or contour indicates the method used to obtain the value. All trend lines were obtained by linearly interleaving the values at time points 4–26 h.

**Table 1 ijms-24-11779-t001:** Growth metrics of *E. coli* strains grown in the presence of methionine and ncAAs.

	*E. coli* BL-21				*E. coli* B834			
	Met	pMet	Aha	Hpg	Met	pMet	Aha	Hpg
r	0.82	0.47	0.52	-	0.63	-	-	-
K	1.79	2.00	1.97	-	1.80	-	-	-
t_5%K_ [h]	4.2	7.1	6.6	-	3.3	-	16.1 *	-
t_1%r_ [h]	13.3	23.3	21.0	-	15.2	-	>25.0	-

r—unrestricted growth rate; K—maximum population size; t_5%K_—time required for the culture to reach 5% of its maximum population size; t_1%r_—time required for the culture to reach stationary phase expressed as 1% of its unrestricted growth rate; -—not calculated; * 5% K of *E. coli* BL-21 grown in M9 medium with Aha used.

**Table 2 ijms-24-11779-t002:** Methionine site-containing peptides with high propensity for ionization.

Protein	Methionine Sites in		Sequence	From-To	Monoisotopic Mass (Da)
	Protein	Peptide			
b_5_M46	1	1	K.FLEEHPMGEEVLR.E	39–51	1584.7606
MBP-GFP	12	1	K.SAMPEGYIQER.T	494–504	1279.5867
		2	K.GEIMPNIPQMSAFWYAVR.T	327–344	2109.0176
pspA	7	1	R.QQALMLR.H	136–142	858.4746
		2	R.LMIQEMEDTLVEVR.S	31–44	1704.8426

From-To—range of amino acids within the protein sequence; dots in sequences denote peptide cleavage sites.

**Table 3 ijms-24-11779-t003:** Amino acids and their derivatives found at methionine sites of the peptides.

Amino Acid	Side Chain	Formula	Monoisotopic Mass (Da)	Δ Mass (AA-Met) (Da)	Origin of the Derivative
Methionine	-(CH_2_)_2_-S-CH_3_	C_5_H_11_NO_2_S	149.0510	-	-
MetOx	-(CH_2_)_2_-S(=O)-CH_3_	C_5_H_11_NO_3_S	165.0460	15.9950	sp
pMet	-(CH_2_)_2_-C(=N_2_)-CH_3_	C_6_H_11_N_3_O_2_	157.0851	8.0341	-
pMet D1	-(CH_2_)_2_-CH=CH_2_	C_6_H_11_NO_2_	129.0790	−19.9720	pi
	-CH_2_-CH=CH-CH_3_				
Aha	-(CH_2_)_2_-N=N^+^=N^−^	C_4_H_8_N_4_O_2_	144.0647	−4.9863	-
Aha D1	-(CH_2_)_2_-NH_2_	C_4_H_10_N_2_O_2_	118.0742	−30.9768	bc/sp
Aha D2	-(CH_2_)_2_-OH	C_4_H_9_NO_3_	119.0582	−29.9928	bc/sp
Aha D3	-CH=CH_2_	C_4_H_7_NO_2_	101.0477	−48.0033	bc/sp
Hpg	-(CH_2_)_2_-C≡CH	C_6_H_9_NO_2_	127.0633	−21.9877	-

Δ Mass (AA-Met)—mass difference of the respective amino acid and methionine; MetOx—methionine sulfoxide, D1-3—derivatives of the respective amino acid; bc—bacteria culturing, sp—sample processing, pi—peptide ionization by MALDI.

**Table 4 ijms-24-11779-t004:** Differences in the determination of the aggregated incorporation of ncAA and its derivatives in MBP-GFP by MS and LCMS.

ncAA	pMet (%)				Aha (%)				Hpg (%)			
Approach	MS		LC-MS		MS		LC-MS		MS		LC-MS	
Peptide	P1	P2	P1	P2	P1	P2	P1	P2	P1	P2	P1	P2
Ave _(P1/2)_	56.8	54.2	74.1	73.0	64.6	73.3	53.0	57.0	46.6	59.7	51.4	58.9
S.D. _(P1/2)_	5.4	3.6	1.5	0.9	2.6	2.1	2.7	3.1	7.3	6.1	4.1	5.9
Ave _(total)_	55.5		73.5		68.9		55.0		53.1		54.7	
S.D. _(total)_	4.8		1.3		5.0		3.5		9.4		6.2	
Δ Ave _(P1-P2)_	2.7		1.1		−8.7		−3.9		−13.1		−7.5	
Δ Ave _(MS-LCMS)_			−17.3	−18.8			11.6	16.4			−4.8	0.8

i.r.—aggregated incorporation rate of ncAA and its derivatives; peptides SAMPEGYIQER (P1) and MEYNYNAHNVYIMTDK (P2); Ave _(P1/2)_, S.D. _(P1/2)_—average i.r. at peptide P1 or P2, and the respective standard deviation; Ave _(total)_, S.D. _(total)_—average i.r. at both P1 and P2 peptides, and the respective standard deviation; Δ Ave _(P1-P2)_—absolute difference in average i.r. between peptides P1 and P2, Δ Ave _(MS-LCMS)_—absolute difference in average i.r. at peptide P1 or P2 determined by MS and LC-MS; i.r. determined at 4–21 h time points were averaged.

## Data Availability

Data not available.

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
