# Peer review of "Photo-Methionine, Azidohomoalanine and Homopropargylglycine Are Incorporated into Newly Synthesized Proteins at Different Rates and Differentially Affect the Growth and Protein Expression Levels of Auxotrophic and Prototrophic E. coli in Minimal Medium"

_ijms, 2023, doi:10.3390/ijms241411779_

Round 1

Reviewer 1 Report

In this paper, the authors examined the optimal conditions for the highly efficient incorporation of three ncAAs, pMet, Aha, and Hpg, which have a structure similar to methionine, into proteins.

 Unfortunately, the study seems to be not of much interest to general readers.

 However, the experimental data have been discussed in detail and will provide useful information for researchers conducting closely related studies.

 Major points:

 1.     Line 259: The data supporting the identification of the extra induced band as PspA must be presented.

2.     In B834, the authors claim that the reason for the poor growth in minimal medium supplemented with ncAA is the lack of trace-level Met supply. I strongly recommend the authors to perform experiments to confirm whether the addition of a small amount of Met, in addition to ncAA, to the medium can restore growth. In particular, this may provide important clues to determine whether the growth inhibition in Hpg can be explained by shortage of Met or by Hpg toxicity.

3.     Figure 5, 6, and Table 4: The meaning of the value of ncAA incorporation is not clear. What was incorporated into the proteins that did not have ncAA incorporation; methionine, the degradation products shown in Table 3, or both? Interpretation of the data should be clearly indicated.

4.     Table 3 was poorly discussed.

 Minor points:

Line 62: “quantity”

Concentration?

Line 146-151: “E. coli BL-21 culture in the presence of Hpg did not exceed OD600 ~0.1 during the course of the experiment indicating its strong influence on cell metabolism, which, unlike the effect observed with the other 2 ncAAs, either cannot be countered by the increase in methionine concentration by de novo synthesis or methionine concentration stays low because the biosynthesis itself is being blocked.”

Since growth was only examined at a single concentration of ncAA, it should be carefully determined if the action is a qualitatively different.

Line 169-171: “In sharp contrast, the presence of Hpg in the medium had a greater effect on the cells, where methionine biosynthesis must have been inhibited or their metabolism severely affected in some other way in E. coli prototrophic strain, as the character of its growth did not differ significantly from that of auxotrophic strain in the presence of the ncAAs.”

The statement in this section appears to be contradictory to the definition in INTRODUCTION that ncAA does not injure cells.

Line 258: Figure 3E,F

Figure 3C,F maybe correct.

Line 259: Figure 3C

Figure 3E?

Line 288-289: Figure S2

Method S2?

Figure 1, Table 1:

In Fig1, it appears difficult to calculate t1%r for the B834 strain. In Fig1, the scale of the vertical axis should be adjusted and presented so that the growth curve of BL21 (Hpr) and B834 (pMet, Aha, Hpr) can be clearly recognized.

Figure 5 Inkorp.

Incorporation?

Table 4 (Values)

% ?

//

Reviewer 2 Report

Peer-review of the manuscript entitled "Photo-methionine, Azidohomoalanine, and Homopropargylglycine have distinct effect on the growth of auxotrophic and prototrophic E. coli in minimal medium, alter protein expression levels and incorporate into newly synthesized proteins at different rates." by Jecmen, T., Tuzhilkin, R., and Sulc, M.

In this manuscript, the authors focused on three non-canonical amino acids (ncAAs), photo-methionine (pMet), azidohomoalanine (Aha), and homopropargylglycine (Hpg). Their overall goal was to find how ncAAs would affect the growth of prototrophic E. coli strain BL-21 and auxotrophic E. coli strain B834. Using these strains, the authors examined the expression of two recombinant proteins, b5M46 and MBP-GFP, and determined the percentage of methionine substitutions with ncAAs.

This research has merit and could serve as a reference for researchers looking to optimize the incorporation of ncAAs into recombinant proteins expressed in prokaryotes. Below, please find our comments:

1.     The manuscript's result section is long and contain unnecessary details which distract from this study's main objectives and accomplishments. It is advised that the authors provide a clear and concise sentence at the beginning of each results subsection, summarizing the main findings and the implications that those results have on the overall study.

2.     The heading for section 2.1 ("Bacterial Growth in the Presence of Non-canonical Amino Acids") is very similar to the header for section 2.2 ("Bacterial Growth in the Presence of Non-canonical Amino Acids during Protein Expression.”) These sections should be given unique titles.

3.     Page 3, line 97 states that the research is "comparing expression of two different recombinant proteins…" and line 107 says, "We focused on three proteins." This difference in the number of proteins studied should be explicitly explained, and the name/function of the proteins should be included in the introduction.

4.     Figure 3 on pg. 6 should contain a control lane for uninduced cells.

5.     The authors describe the use of MS and LCMS to determine the incorporation of ncAAs into recombinant proteins. Table 4 on page 11 details the data collected from these analytical tests. However, the graphs generated by these analyses are not provided in the supplementary information. Whether these graphs need to be included should be at the editor's discretion.

6.     Lane 59: “Because aminoacyl-tRNA is subsequently recognized by a ribosome strictly on the basis of anticodon sequence…” implies that once aminoacylation is completed by aminoacyl-tRNA synthetases, aminoacyl-tRNAs are used by the ribosome at the same rate regardless of the nature of the amino acids they carry. In the publication entitled “Uniform binding of aminoacyl-tRNAs to elongation factor Tu by thermodynamic compensation” (Science, 2001), Lariviere and collaborator have unambiguously demonstrated that elongation facture EF-Tu actually discriminates against non-canonical amino acid / tRNA pairs. I invite Jecmen and collaborators to revise their statement.

7.     The manuscript contains run-on sentences that should be edited. Examples include “As our current results suggest…” line 90 and “The latter was probably due…” line 143.

.

Round 2

Reviewer 1 Report

Moreover, the main aim of this study is expression of proteins with high content of ncAAs and we were able to achieve positive results by transferring culture pre-grown in nutritionally rich medium to minimal medium where protein expression is induced. It is also worth mentioning, that of the three tested methionine analogs, the best incorporation and sufficient protein yield was already achieved with Hpg, in prototrophic E. coli.”

If the authors claim that this is the main aim and conclusion of their paper, they should clearly state these in the abstract and at the end of the discussion. They should also show the reader, in plain language, how a protein with a sufficiently high ncAA content can be obtained. In the current manuscript, the authors' aims and conclusions seem to be ambiguous.

//

Author Response

Comment: “Moreover, the main aim of this study is expression of proteins with high content of ncAAs and we were able to achieve positive results by transferring culture pre-grown in nutritionally rich medium to minimal medium where protein expression is induced. It is also worth mentioning, that of the three tested methionine analogs, the best incorporation and sufficient protein yield was already achieved with Hpg, in prototrophic E. coli.”

If the authors claim that this is the main aim and conclusion of their paper, they should clearly state these in the abstract and at the end of the discussion. They should also show the reader, in plain language, how a protein with a sufficiently high ncAA content can be obtained. In the current manuscript, the authors' aims and conclusions seem to be ambiguous.

Reply: Although we believe that the previous version of Abstract already contained the key information about the main purpose (originally line 24), we have also stressed the main goal more explicitly on line 15. We have made only minor changes to the abstract so as not to exceed the word limit.

We have moved Conclusions section summarizing expression of protein with sufficiently high ncAA content from the end of the manuscript to immediately after Discussion (line 507), and have substantially revised the text to make it clearer and more concise.